# Derivation and validation of a non-invasive optoacoustic imaging biomarker for detection of patients with intermittent claudication

## Abstract

**Background** Peripheral arterial disease (PAD) affects more than 200 million people worldwide, with symptoms ranging from none to severe. Despite these different diagnostic options, patients with unclear leg pain remain challenging to diagnose. The primary objective of this study was to evaluate whether multispectral optoacoustic tomography (MSOT) can discriminate between healthy volunteers (HV) and patients with intermittent claudication (IC) by assessing hemoglobin-related biomarkers in calf muscle tissue.

**Method** In this monocentric, cross-sectional, observational diagnostic trial (NCT05373927) $n = 102$ patients were included in two independent derivation (DC, $n = 51$) and validation cohorts (VC, $n = 51$). MSOT was performed before and after standardized heel raise provocation and was compared to standardized PAD diagnostics including pulse palpation, ankle brachial index (ABI), duplex sonography, 6-minute walk test (6MWT), assessment of health-related quality of life (VASCUQOL-6), and angiography (aggregated TransAtlantic Inter-Society Consensus classification, aTASC).

**Results** Here we show that MSOT is capable of differentiating IC and HV with an area under the receiver operator characteristics curve (AUROC) in DC by 0.99 (sensitivity: 100%, specificity: 95.8%) and in the VC by 0.95 (sensitivity: 96.2%, specificity: 96.0%). MSOT-derived oxygenation positively correlates with the ABI post-exercise (R = 0.83, $P = 2.31 \times 10^{-26}$), the absolute walking distance in the 6MWT (R = 0.77, $P = 3.40 \times 10^{-21}$), the VASCUQOL-6 (R = 0.79, $P = 4.82 \times 10^{-23}$) and negatively with aTASC classification (R = -0.80, $P = 2.92 \times 10^{-24}$).

**Conclusions** Post-exercise MSOT-derived saturation in the calf muscle is validated as a non-invasive imaging biomarker to distinguish HV and IC patients yielding high sensitivity and specificity.

## Plain language summary

Many people suffer from peripheral arterial disease (PAD), which can cause insufficient blood supply to the lower legs. However, it is often difficult to distinguish whether symptoms of leg pain are a result of the early stages of PAD or other diseases. We examined 50 people with early-stage PAD and 52 healthy people before and after exercising their lower leg muscles using a molecular-sensitive ultrasound imaging method. People with early-stage PAD had less oxygen in the muscle tissue than healthy people. This imaging approach could provide a fast and reliable tool to assess PAD and be used in routine clinical care to identify people with PAD.

With 237 million cases worldwide, peripheral artery disease (PAD) ranges among the three most common cardiovascular diseases[1]. The occlusion of vessels, as seen in PAD, can progress asymptomatically and finally result in symptoms like intermittent claudication (IC), rest pain or ischemic ulceration[2,3]. First-line diagnostics includes palpation of pulses, measurement of the blood pressure derived ankle-brachial index (ABI), or color-coded duplex sonography. Other non-invasive diagnostic modalities are mainly used in later stages, which are for example transcutaneous oxygen pressure (tcPO2), laser-doppler flowmetry, skin-perfusion pressure of fluorescence angiography[2,3]. However, to confirm the ischemic origin of symptoms, invasive angiographic imaging procedures are often required.

Despite the existence of these various diagnostic options, patients with unclear leg pain during walking remain challenging for diagnosis[4]. In these cases, a diagnostic tool for direct assessment of the affected organ, namely the lower leg musculature in PAD, would be helpful.

✉e-mail: ferdinand.knieling@uk-erlangen.de; ulrich.rother@uk-erlangen.de

In this regard, Multispectral Optoacoustic Tomography (MSOT) enables visualization and quantification of muscle tissue properties, such as hemoglobin of different oxygenation states over a broad range of applications[5–10]. MSOT has already demonstrated feasible to detect differences in terms of hemoglobin saturation of the calf muscle between healthy volunteers (HV) and different stages of PAD[11]. However, its diagnostic accuracy to differentiate between healthy and early claudication stage, which would be clinically relevant, is still elusive.

Therefore, this study aimed to develop appropriate thresholds and subsequently validate the diagnostic performance of MSOT to discriminate HV from IC patients by assessing the calf muscle before and after exercise.

Our study results show that MSOT imaging performed after heel-raise exercise can discriminate patients with intermittent claudication from healthy volunteers with high sensitivity and specificity. Thus, non-invasive optoacoustic imaging provides a rapid, bedside and accurate tool to identify the ischemic origin of leg pain in such patients.

## Patients and Methods

### Trial Overview

The prospective data collection and analysis was designed as a monocentric, cross-sectional, observational diagnostic study and performed at the Department of Vascular Surgery of the University Hospital Erlangen, Germany, between 9th May 2022 and 14th July 2022 (study completion). The trial was approved by the local ethics committee of the University Erlangen-Nürnberg (22-62-Bm) and registered on clinicaltrials.gov (NCT05373927). The clinical study followed adhered to the Declaration of Helsinki. All patients and healthy volunteers gave written informed consent prior to participation. A total of 52 healthy volunteers (HV) and 50 IC patients were recruited. In the derivation cohort (DC), 27 HV were compared to 24 IC patients. HV were required to be at least 50 years old, have palpable foot pulses and a normal ABI (values between 0.9 and 1.4). Exclusion criteria were existing diabetes mellitus, chronic renal insufficiency, and IC symptoms. IC patients were included in Fontaine stage IIa/IIb or Rutherford category 1 to 3. The findings in the DC were applied to an independent validation cohort (VC, 25 HV vs 26 IC) to confirm the findings. Inclusion criteria were patients with manifest PAD in Fontaine stage II or Rutherford categories 1-3 or healthy volunteers, both over 18 years of age, who gave informed consent. Patients with PAD stage I, III and IV according to Fontaine or categories 0, 4, 5 and 6 according to Rutherford or healthy volunteers with diabetes mellitus, chronic renal failure, claudication symptoms, abnormal ABI or non-palpable foot pulses were excluded. Subjects were also excluded if they were under 18 years of age, did not provide written consent, or if the investigator had safety concerns (a person with a physical, mental, or psychiatric condition that, in the opinion of the investigator, would compromise the safety of the person or the quality of the data, making the person ineligible for the study).

### Trial Procedures

Potential participants were screened for inclusion and exclusion criteria and if suitable for the study, they were informed about the procedure, benefits, and risks. After giving written consent, examinations were done (exactly comparable between HV and IC, with except of angiography) during one single appointment within 90 minutes. Anamnesis was taken, in which all essential information related to PAD diagnosis was obtained. The standardized VASCUQOL-6 questionnaire was used to assess the individual PAD-specific quality of life[12]. A standardized non-invasive routine examination for PAD diagnostics was performed, which included pulse palpation, ABI before and after exercise and color-coded duplex ultrasound (CCDS)[3]. The vascular occlusion profile in IC patients was confirmed by angiographic imaging (either magnetic resonance angiography, computed tomography angiography or digital subtraction angiography) (Supplementary Table 1). All MSOT measurements were performed with the same CE-certified system (Acuity Echo, iThera Medical GmbH, Munich) at wavelengths between 700 and 1210 nm. By using the integrated B-mode ultrasound function, the optimal position for imaging the triceps surae muscle was determined in the B-mode scan. The final position of the transducer was marked and the exact same position was retrieved for the pre- and post-exercise measurement. The MSOT parameters deoxygenated hemoglobin (Hb), oxygenated hemoglobin (HbO$_2$) and MSOT saturation (mSO$_2$) were then determined for the muscle (exemplary MSOT images can be found in Fig. 3). The first measurement was taken after a rest of 10 minutes in a lying position (mean of two independent measurements). After this initial measurement, all participants performed the heel raise exercise to strain the calf muscle, repeated until pain or muscle exhaustion (in the healthy group) occurred (Supplementary Fig. 1). Directly after the exercise, another ABI measurement and the second MSOT measurement at the marked position of the calf muscle were performed (mean of three independent measurements within 150–250 s).

After a resting phase of at least 30 min, the participants underwent a Six-Minute Walk Test (6MWT) to determine the relative (distance until the first pain occurs), absolute (distance until the first stop due to calf pain) and total (distance walked in six minutes) walking distance.

### Survey of study relevant medical history

To collect relevant medical information from the study participants, the electronic patient files and the collected information during anamnesis were used. In addition to general information, such as date of birth, sex, weight, height and skin type according to Fitzpatrick, vascular risk factors (smoking, arterial hypertension, dyslipidemia, diabetes mellitus, obesity with BMI > 30 kg/m$^2$, positive family history) were identified. Previous performed vascular surgeries (interventional/open surgery/both), as well as important previous diseases, such as coronary artery disease, heart failure, atrial fibrillation, myocardial infarction, preterminal and terminal chronic kidney disease (with indication of the creatinine value), carotid stenosis and stroke were also determined. Furthermore, the use of relevant medications (anti-hypertensives, lipid-lowering agents, antidiabetics, acetylsalicylic acid, clopidogrel/ticagrelor, heparin, oral anticoagulation, coumarin, naftidrofuryl, cilostazol, and prostanoids) was queried. Finally, the participants were asked about their subjective assessment of how far they could walk until the first pain occurred (relative walking distance), or until they must stop for the first time due to pain (absolute walking distance).

### Heel raise exercise

The heel raise exercise is designed to achieve maximal load and exhaustion of the calf muscle. For this purpose, the subject is asked to change from the normal stance to the toe-ball stance for at least 30 s at 1-s intervals. If there is no pain and no fatigue after 30 s in calf muscle, the subject should continue to alternate between normal stance and toe-ball stance until either event occurs. Supplementary Fig. 5 shows a schematic illustration of the heel raise exercise.

### Data Analysis

The examiners for the final MSOT data analysis were blinded to the results of the clinical assessments. All data processing for measurements of imaging parameters before and after the heel raise exercise was performed by iLabs software (Version 1.2.9, iThera Medical GmbH, Munich). As demonstrated before, deoxygenated hemoglobin (Hb), oxygenated hemoglobin (HbO$_2$) and MSOT-derived oxygenation (mSO$_2$, calculated: HbO$_2$/(Hb+HbO$_2$)) were assessed after drawing a standardized region of interest (ROI) directly below the muscle fascia of the triceps surae muscle[11]. The Hb and HbO$_2$ values were obtained directly from the generated data using an algorithm, while mSO$_2$ equals the quotient of HbO$_2$ by Hb+HbO$_2$. It should be emphasized that the parameter described as mSO$_2$ does not correspond to the actual physical oxygen saturation and should therefore be considered as an independently generated parameter. The other MSOT values are given in arbitrary units (a.u.).

Angiographic images of the IC patients were reviewed by two independent investigators (M.C., J.K.) and categorized according to the transatlantic inter-society consensus document II (TASC)[13,14]. In the event of uncertainties, a third opinion (U.R.) was consulted for a collaborative decision-making. The classification evaluates the anatomic distribution and

**Fig. 1 | Flow chart of patient recruitment and data collection process.** Created in BioRender. Knieling, F. (2025) https://BioRender.com/m34t768.

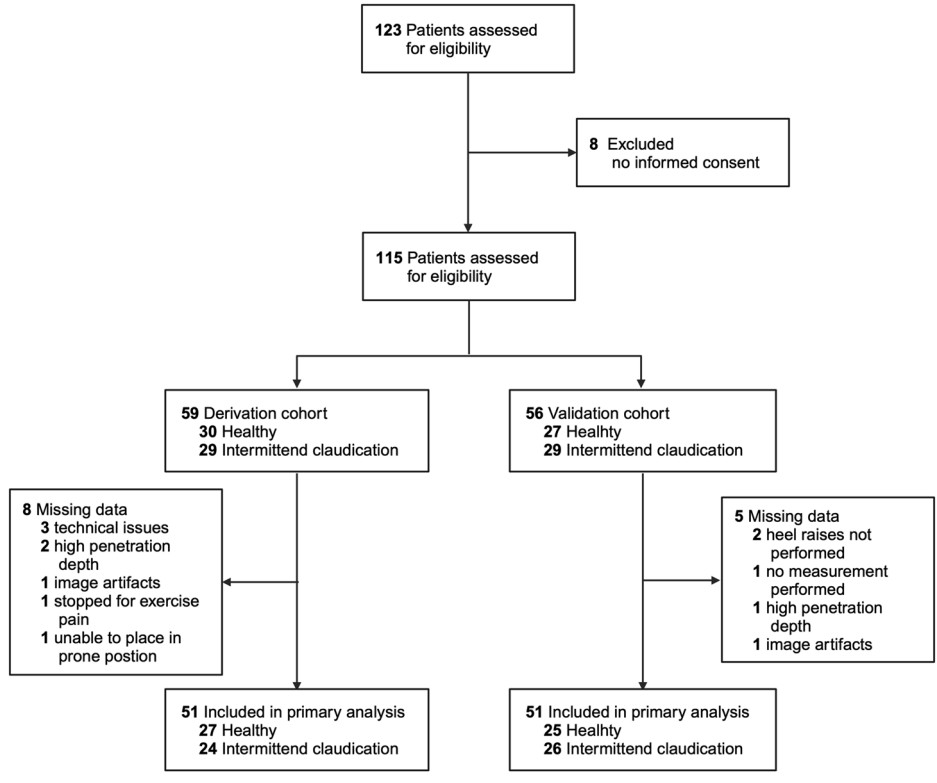

extent of lower extremity vascular lesions at three separate levels: Aorto-iliac, femoral popliteal and infrapopliteal. To obtain an overview of the functional severity of the PAD, we used an additional classification (aTASC), which combines the separately reviewed areas and considers the collateralization situation. The additional classification includes three classes (aTASC1: angiographically healthy, aTASC 2: good collateralization capability, aTASC 3 poor collateralization capability)[14–16].

### Statistics and reproducibility
**Sample size.** Statistical sample size calculation based on the results of the previous study (MSOT_PAD, NCT04641091) suggests a necessary study collective of at least 48 to a maximum of 124 subjects/patients. The two-sample t-test power calculation with a power of 0.8 gave an n of about 12 for each group (study group 1 - healthy volunteers, study group 1—claudication patients, study group 2—healthy volunteers, study group 2—claudication patients). Additional sample size planning based on the AUC, which should be at least 0.7 for a relevant significance, gave an n of 31 subjects/patients for each group (with a significance level of 0.05 and a power of 0.8). These calculations were based on data comparing healthy subjects with patients in stage IIb who, however, had not been adequately exercised (all only with the same walking distance of 150 meters). Due to the hypothesis of the study that the ability to differentiate between healthy subjects and claudication patients can be increased by adequate provocation testing, the sample size estimated here based on the previous data is to be regarded as conservative. For this reason, after the inclusion of the first 25 patients and the first 25 healthy test persons, the necessary total sample was reevaluated. During this interims analysis we found, a minimum sample size of 11 (oxygenated hemoglobin [$HbO_2$])-33 (deoxygenated hemoglobin [Hb]) participants in each clinical group (HV, IC) (80% power, alpha of 0.05, two-sample t-test power calculation, two-sided).

### Statistical analyses
Continuous variables are given as mean values with standard deviation. The values determined correspond to the average of seven neighboring frames (technical replicates) and represent the mean of two (pre) or three independent measures (post, biological replicates).

Categorical factors are presented as absolute and relative frequencies. Correlations are given by Pearson's correlation coefficient (R). Single missing values led to exclusion of subjects for the complete case analysis. In case no muscle was appropriately imaged in the MSOT frames, the subject was excluded from all further analyses. The threshold for statistical significance was set to a p-value of <0.05. When interpreting differences between groups, the p-value, effect size and overall consistency of the results were considered. Analyses were performed using IBM SPSS Statistics (version 28, IBM Corp., N.Y., USA), R Statistics (version 4.0.5.) and GraphPad PRISM (version 10.4.1, GraphPad Software, USA). Statistical differences of DC and VC were tested by unpaired t-tests and matched comparison by paired t-tests. To illustrate the diagnostic quality to differentiate between HV and IC by MSOT, the sensitivity, specificity, area under the curve (AUC) and the cut-off point (derived from the optimal Youden index) were determined in the DC. The cut-off point was then applied to the VC to validate the results from the DC. Estimates were accompanied by 95% confidence limits (95% CIs).

### Reporting summary
Further information on research design is available in the Nature Portfolio Reporting Summary linked to this article.

## Results
### Patient characteristics
A total of 123 individuals were consecutively screened for enrollment between 9th May 2022 and 14th July 2022, of whom 115 gave written informed consent. In 109 cases, the study was completed with a full data set and finally 102 of them were used for data processing (compare study flow chart, Fig. 1). HV had a mean age 63.8 ± 8.0 years and 45.1% were female. Patients with IC were 60.0 ± 6.0 years of age and 42.3% were female. A description of the patient characteristics is given in Table 1. We found no difference between the characteristics of the consecutively collected separated DC and VC (Supplementary Table 2).

## Table 1 | Patient characteristics

| Characteristics | Eligible patients | Patients in primary analysis | HV | IC |
|---|---|---|---|---|
| | (*n* = 115) | (*n* = 102) | (*n* = 52) | (*n* = 50) |
| Age, yrs. (mean, SD) | 64.4 ± 8.3 | 63.8 ± 8.0 | 60.0 ± 6.0 | 67.8 ± 7.9 |
| **Sex, *n* (%)** | | | | |
| Female | 54 (47) | 46 (45.1) | 30 (57.7) | 16 (32.0) |
| **Risk factors for PAD, *n* (%)** | | | | |
| Ever Smoking | 62 (53.9) | 55 (53.9) | 8 (15.4) | 47 (94.0) |
| Arterial hypertension | 60 (52.2) | 56 (54.9) | 11 (21.2) | 45 (90.0) |
| Dyslipidemia | 53 (46.1) | 47 (46.1) | 17 (32.7) | 30 (60.0) |
| Diabetes mellitus | 17 (14.8) | 16 (15.7) | 0 (0) | 16 (32.0) |
| Obesity (BMI > 30) | 15 (13.0) | 12 (11.8) | 1 (1.9) | 11 (22.0) |
| **Relevant diseases, *n* (%)** | | | | |
| Coronary artery disease | 18 (15.7) | 17 (16.7) | 0 (0) | 17 (34.0) |
| Carotid artery stenosis | 21 (18.3) | 21 (20.6) | 2 (3.8) | 19 (38.0) |
| History of myocardial infarction | 8 (7.0) | 8 (7.8) | 0 (0) | 8 (16.0) |
| Stroke | 8 (7.0) | 8 (7.8) | 1 (1.9) | 7 (14.0) |
| **Current medication, *n* (%)** | | | | |
| Antihypertensive | 58 (50.7) | 54 (52.9) | 11 (21.2) | 43 (86.0) |
| Lipid-lowering agent | 41 (35.7) | 36 (35.3) | 4 (7.7) | 32 (64.0) |
| Antidiabetic | 15 (13.0) | 14 (13.7) | 0 (0) | 14 (28.0) |
| Previous revascularization procedure, *n* (%) | 26 (22.6) | 22 (21.6) | 0 (0) | 22 (44.0) |
| **Ankle-brachial-index** | | | | |
| Before the exercise | 0.83 ± 0.33 | 0.84 ± 0.32 | 1.12 ± 0.06 | 0.56 ± 0.22 |
| After the exercise | 0.70 ± 0.42 | 0.72 ± 0.42 | 1.08 ± 0.10 | 0.33 ± 0.22 |
| Duration of heel raises exercises [mean, seconds] | 108.1 ± 49.5 | 107.6 ± 47.9 | 129.9 ± 52.2 | 84.4 ± 28.5 |
| VASCUQOL-6 score | 18.58 ± 6.23 | 18.65 ± 6.20 | 24.00 ± 0.00 | 13.08 ± 4.16 |
| **Walking distance in 6MWT [m]** | | | | |
| Relative | 118 ± 71 | 115 ± 71 | N/A | 115 ± 71 |
| Absolute | 211 ± 92 | 207 ± 88 | N/A | 207 ± 88 |
| Total | 460 ± 137 | 463 ± 142 | 581 ± 43 | 340 ± 98 |
| **PAD stage according to Fontaine, *n* (%)** | | | | |
| HV | 57 (51.0) | 52 (51.0) | 52 (100) | 0 (0) |
| IIa | 28 (24.3) | 24 (23.5) | 0 (0) | 24 (48.0) |
| IIb | 30 (26.1) | 26 (25.5) | 0 (0) | 26 (52.0) |
| **PAD stage according to aTASC, *n* (%)** | | | | |
| aTASC 1 | 58 (50.4) | 53 (52.0) | 52 (100) | 1 (2.0) |
| aTASC 2 | 47 (40.9) | 41 (40.2) | 0 (0) | 41 (82.0) |
| aTASC 3 | 10 (8.7) | 8 (7.8) | 0 (0) | 8 (16.0) |

Categorical data represented by absolute and relative frequencies, continuous data by mean ± standard derivation. Statistical comparison of the HV and IC and total cohort using the Chi-squared test for categorical data and the t-test for continuous data. (HV, healthy volunteers; IC, patients with peripheral arterial disease with in Fontaine stage IIa/IIb or Rutherford category 1 to 3; VASCUQOL-6, Vascular Quality of Life Questionnaire-6; 6MWT, Six-Minute-Walk-Test; PAD, peripheral arterial disease; aTASC (Type 1: HV or IC with no signs of stenosis or occlusion in angiography; Type 2: signs of stenosis or occlusion in the femoropopliteal and/or infrapopliteal area or TASC-II-level A or B in the aortoiliac area; Type 3: TASC-II-level C or D in the aortoiliac area); N/A, not applicable).

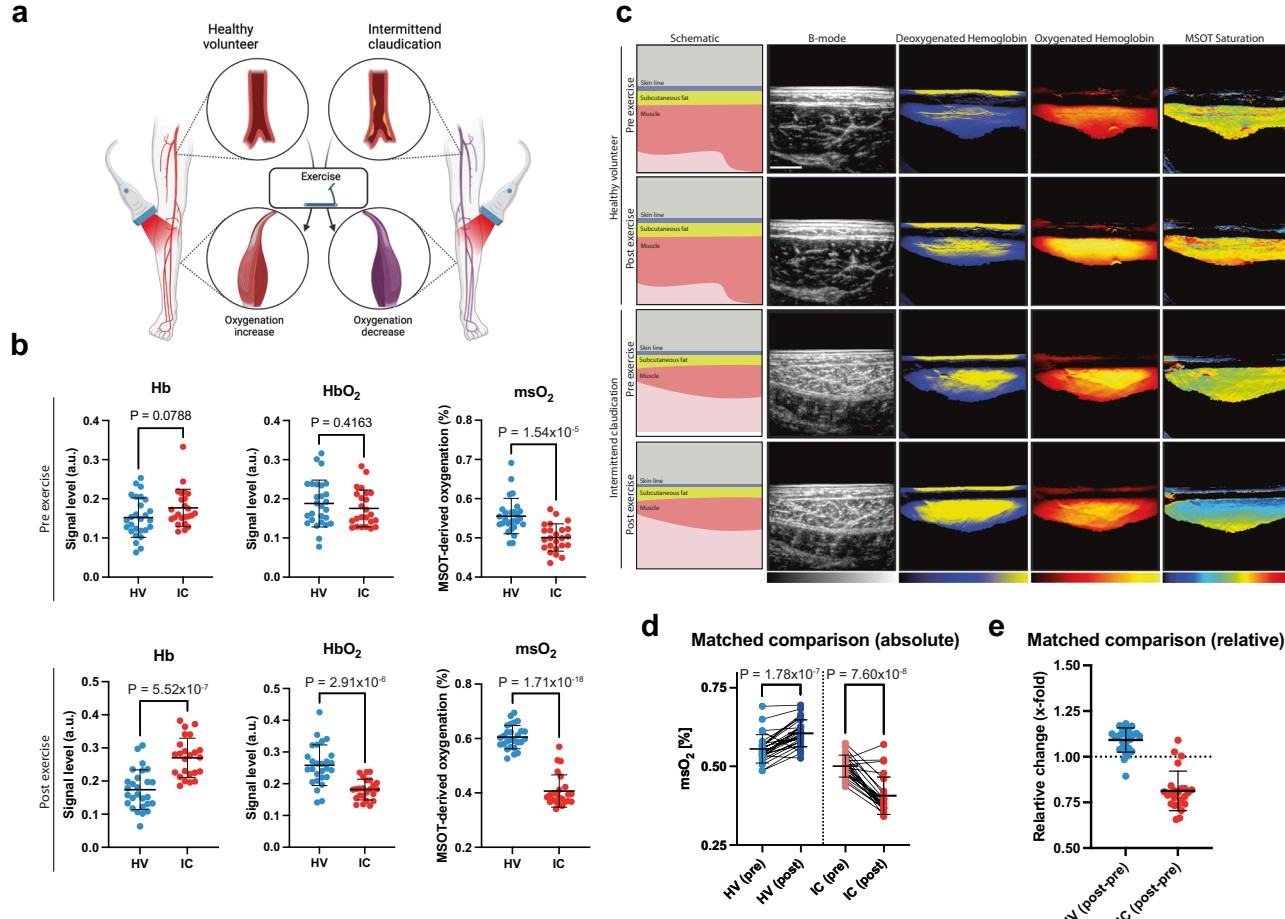

**Fig. 2 | MSOT imaging results before and after the heel raise exercise. a** MSOT imaging approach in the study. Created in BioRender. Knieling, F. (2025) https://BioRender.com/p04l195. **b** Dot plots of all patients in the derivation cohort pre- and post-exercise (for all parameters: HV $n = 27$, IC $n = 24$). **c** MSOT images of a healthy volunteer (HV) and a patient with peripheral arterial disease in Fontaine stage IIa/IIb or Rutherford category 1 to 3 (IC) before (Pre) and after (Post) the heel raise exercise showing the parameters deoxygenated hemoglobin (Hb), oxygenated hemoglobin (HbO2) and MSOT saturation (mSO2). The first column shows a schematic representation of the visible anatomical structures (skin line, sub-cutaneous fat, muscle); the second shows the sonographic image in B-mode. The following columns illustrate the MSOT data of Hb, HbO2 and mSO2, visually represented as a heat map, with the colors reflecting the signal intensity (Hb: dark blue, minimum signal; yellow, maximum signal; HbO2: dark red, minimum signal; yellow, maximum signal; mSO2: dark blue, minimum signal; dark red, maximum signal). **d** Individual comparison of HV (HV (pre) $n = 27$; HV (post) $n = 27$) and IC (IC (pre) $n = 24$; IC (post) $n = 24$) patients using absolute and (**e**) relative mSO2 measurements of post-pre (HV (pre) $n = 27$; HV (post-pre) $n = 27$) and IC (IC (pre) $n = 24$; IC (post-pre) $n = 24$) values. Lines connect individual patients. Error bars show mean ± SD.

## Diametral response of MSOT-derived parameters in HV and IC

All patients underwent standardized MSOT imaging before and after exercise (Fig. 2a). At resting state in the DC, we found no difference between the HV and IC group for the parameters Hb ($0.15 \pm 0.05$a.u. vs. $0.18 \pm 0.05$a.u; $P = 0.0788$) and HbO2 ($0.19 \pm 0.06$a.u. vs. $0.18 \pm 0.05$a.u.; $P = 0.4163$). A substantial difference was observed for mSO2 ($0.56 \pm 0.04$a.u vs. $0.50 \pm 0.04$a.u.; $P = 1.54 \times 10^{-5}$). After heel raise exercise, Hb increased in IC patients ($0.17 \pm 0.06$a.u. vs. $0.27 \pm 0.06$a.u.; $P = 5.52 \times 10^{-7}$). Opposite to Hb, HbO2 increased in the HV ($0.26 \pm 0.06$a.u. vs $0.182 \pm 0.03$a.u.; $P = 2.91 \times 10^{-6}$). Given this disease-dependent diametric response, mSO2 logically also differentiated IC and HV group ($0.41 \pm 0.06$a.u. vs. $0.61 \pm 0.04$a.u.; $P = 2.91 \times 10^{-6}$) (Fig. 2b). Exemplary MSOT images are given in Fig. 2c. It is possible to distinguish between HC and IC patients (Fig. 2d, e), as the MSOT muscle measurements show diametrically different behavior after stress.

## Diagnostic properties of MSOT to distinguish HV and IC

To assess the diagnostic properties of MSOT to distinguish HV and IC we calculated the receiver operating characteristics (ROC). At resting state, the corresponding area-under-the-curve (AUC) for Hb was 0.65 (95%CI 0.49;0.80), for HbO2 was 0.58 (95%CI 0.42; 0.74), and was mSO2 0.85 (95%CI 0.74;0.96) (Fig. 3a–c). After heel raise exercise, the AUC of Hb increased to 0.87 (95%CI 0.77;0.97). The associated sensitivity in the DC was 74.1% (95%CI 55.3–86.8%) and the specificity 91.7% (95%CI 74.2–98.5%). Similarly, we found a sensitivity of 76.9% (95%CI 58.0–89.0%) and a specificity of 72.0% (95%CI 52.4–85.7%) in the VC (cut-off: 0.20a.u.) (Fig. 3d). Opposite to Hb, were HbO2 increased in the HVs, the AUC was 0.87 (95%CI 0.76;0.97), with a sensitivity of 66.7% (95%CI 47.8–81.4%) and a specificity 100% (95%CI 86.2–100%) in the DC. This was validated with a sensitivity of 100% (95%CI 87.1–100%) and a specificity of 72.0% (95%CI 52.4–85.7%) (cut-off: 0.24a.u.) in the VC (Fig. 3e). The quotient of the two previous parameters, mSO2 had an area under ROC curve of 0.99 (95%CI 0.97;1.00), with a sensitivity of 100% (95%CI 87.5-100%) and a specificity of 95.8% (95%CI 79.8–99.8%) (cut-off: 0.52a.u.). In the VC, a sensitivity of 96.2% (95%CI 81.1–99.8%) and a specificity of 96.0% (95%CI 80.5-99.8%) was confirmed (Fig. 3f). A complete overview divided in DC and VC showing the described parameters before and after the exercise is given Supplementary Table 3. It can be shown that the saturation values in the muscle measured with MSOT are independent of the amount of hemoglobin present in the blood of the participants (Supplementary Fig. 2). In addition,

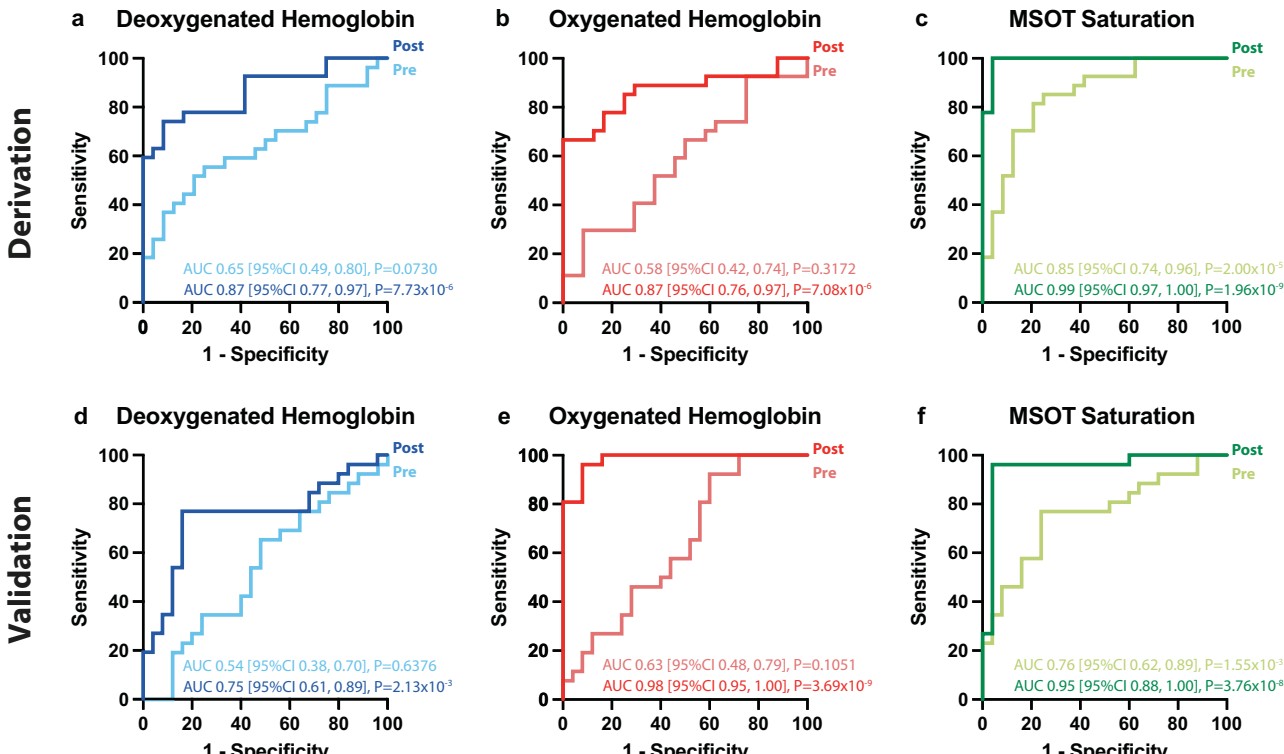

**Fig. 3 | Receiver operating characteristic analysis of MSOT parameters to differentiate healthy volunteers from patients with intermittent claudication.**
**a** Receiver operating characteristic (ROC) analysis comparing the area under the curve (AUC) of the Multispectral Optoacoustic Tomography (MSOT) parameters deoxygenated hemoglobin (Hb, blue curves), **b** oxygenated hemoglobin (HbO$_2$, red curves) and **c** MSOT saturation (mSO$_2$, green curves) of patients with peripheral arterial disease in Fontaine stage IIa/IIb or Rutherford category 1 to 3 (intermittent claudication, IC) and healthy volunteers (HV) before (Pre, light curve) and after (Post, dark curve) the heel raise exercise in the derivation cohort (DC). **d** Hb, **e** HbO$_2$, **f** mSO$_2$ of the validation cohort (VC) pre-exercise (light color curve) and post-exercise (dark color curve).

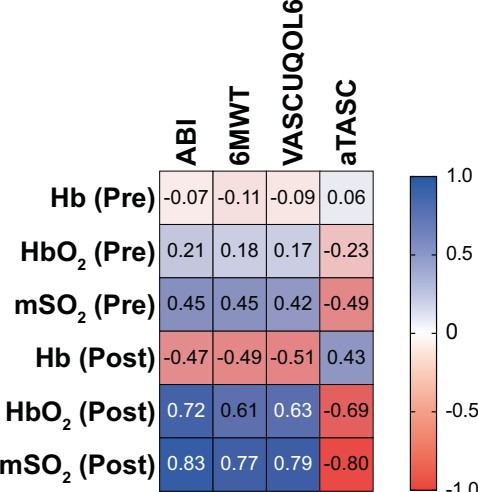

**Fig. 4 | Correlation of MSOT parameters with clinical assessments.** Correlation matrix of the MSOT parameters deoxygenated hemoglobin (Hb), oxygenated hemoglobin (HbO$_2$), MSOT saturation (mSO$_2$) before (Pre) and after (Post) heel raise exercise with ankle-brachial-index after heel raise exercise (ABI), the absolute walking distance in the Six-Minute Walk Test (6MWT), the score in the Vascular Quality of Life Questionnaire-6 (VASCUQOL-6), the score in the aggregated Trans-Atlantic Inter-Society Consensus Document on Management of Peripheral Arterial Disease classification II (aTASC). Correlation given by Pearson correlation coefficient (R) are indicated as strongly positive (dark blue) and strongly negative (dark red).

although there are differences between women and men, mSO$_2$ in particular —both in HV and IC—is comparable (Supplementary Fig. 3).

**Correlation, agreement and bias with clinical standard assessments**

Using the merged cohort (DC and VC, $n = 102$, with except of aTASC, where only IC patients were assessed), the correlation between MSOT parameters (Hb, HbO$_2$ and msO$_2$) before and after the heel raise exercise were analyzed (Fig. 4). At resting state, Hb, HbO$_2$ and mSO$_2$ showed low correlation to standard assessments (range of R ABI: -0.07 to 0.45, 6MWT: -0.11 to 0.45, VASCUQOL-6: -0.09 to 042, aTASC: 0.06 to -0.49).

After exercise, Hb negatively correlated with ABI (R = -0.47, 95%CI -0.61; -0.30, $P = 9.25 \times 10^{-7}$), 6MWT (R = -0.5095%CI -0.63;-0.33, $P = 1.26 \times 10^{-7}$), VASCUQOL-6 (R = -0.5195%CI -0.64;-0.35, $P = 5.04 \times 10^{-8}$) and positively to aTASC (R = 0.43, 95%CI 0.25;0.57, P = 7.54$\times 10^{-6}$). By contrast, HbO$_2$ positively correlated with ABI (R = 0.72, 95%CI 0.61;0.80, $P = 2.78 \times 10^{-17}$), 6MWT (R = 0.61, 95%CI 0.47;0.72, P = 8.94$\times 10^{-12}$), VASCUQOL-6 (R = 0.63, 95%CI 0.49;0.73, $P = 1.69 \times 10^{-12}$) and negatively to aTASC (R = -0.69, 95%CI -0.78;-0.57, $P = 1.18 \times 10^{-15}$), the latter correlation analysis restricted to the IC subsample. Finally, mSO$_2$ showed a strong positive correlation with the ABI after the exercise (R = 0.83, 95%CI 0.75;0.88, $P = 2.31 \times 10^{-26}$), the absolute walking distance in the 6MWT (R = 0.77, 95%CI 0.68; 0.84, $P = 3.40 \times 10^{-21}$), the VASCUQOL-6 questionnaire (R = 0.79, 95%CI 0.70; 0.85, 4.82$\times 10^{-23}$) and negative correlation with angiographic findings, stratified according to the aggregated TASC classification (R = -0.80, 95%CI -0.86;-0.72, $P = 2.92 \times 10^{-24}$).

**Adverse events**

No adverse events were observed.

## Discussion

In this study, three different hemoglobin parameters were derived with MSOT in the triceps surae muscle: Hb, $HbO_2$ and $mSO_2$. Using an easy to implement heel raise exercise, diverging MSOT signal responses in the HV and IC groups provided a strong differentiability between healthy volunteers and diseased patients. While diseased patients showed an increase in Hb and a decrease in $HbO_2$, healthy volunteers showed the exact opposite reaction. This is why, the parameter $mSO_2$, which is calculated as the quotient of $HbO_2$ and $Hb+HbO_2$, provided a strong differentiator for both groups. The data was generated in a derivation cohort and verified using an independent validation cohort.

Since the photoacoustic effect has been described by Alexander Graham Bell in the late 19th century[17], multiple developments transferred this technique to modern clinical diagnostic tools like MSOT[18]. Similar technologies have already been used in clinical applications such as Crohn's disease[19,20] or neuromuscular disorders[6,21]. Furthermore, we already demonstrated its applicability for PAD diagnostic to grade different stages of the disease on based on different levels of calf muscle oxygenation[11]. In this first initial trial, the diagnostic accuracy to discriminate between HV and IC patients was modest. The current increase of sensitivity and specificity could be achieved by examining responses to an easy-to-perform exercise-induced challenge prior to imaging.

Heel raise exercises have capability to lead to an exhaustion in the calf muscle even in healthy subjects, while this cannot commonly be reached by simple treadmill exercise. This challenge has proven its usability in PAD patients, as 30 s of heel raise exercise showed comparable post-exercise ABI results to 1 min treadmill walking tests (4 km/h, gradient of 10 degree)[22]. A further advantage of the heel raise exercise is, that in a future perspective, this exercise can easily be implemented even in the outpatient setting to assess IC patients. MSOT muscle imaging is of certain interest, as it enables the direct assessment of endo-organ involvement in patients with intermittent claudication. The current diagnostic procedures in such diseases with various origins are mainly driven by treadmill or the 6MWT, assessing the pre- and post-exercise ABI[3,23]. However, these tests do not objectify the degree of muscle impairment and are often unable to give conclusive decision about the origin of the symptoms. Additionally, ABI is flawed with the known problems in diabetics and patients with chronic kidney disease, by which MSOT is not affected, as shown in the previous study[11].

Audonnet et al. used transcutaneous oxygenation measurements before and after treadmill exercise to distinguished claudication from lower back pain[24]. However, the diagnostic potential of this method in this indication remains limited as it can merely investigate the skin oxygenation. This is, however, indirectly influenced by a competitive flow due to increase need of the muscle. MSOT has the capability to investigate the muscle directly in different locations of the body up to a depth of three centimeters and might therefore help in cases of unclear diagnosis due to overlapping symptoms.

The clinical usability of the MSOT investigation might be predominantly in patients with IC. In this patient cohort, there are often patients with concomitant diseases, especially lumbar ischialgia, hip disorders or even in buttock claudication in which differential diagnosis can be challenging[4]. In this context, the correlation of the MSOT derived values and the results of the 6MWT as well as the health-related quality of life, assessed by the VASCUQUOL-6, seem to be especially interesting, as these results underline the potential of the method in IC patients[25]. Therefore, MSOT might be proposed as an adjunct measurement in patients with IC, as it can facilitate bedside diagnosis in distinct patients.

This study has several limitations. Even though the study contains a derivation and validation cohort, it was conducted in a monocentric setting. Therefore, the results require confirmation in a multicenter setting to address reproducibility. With regard to technical limitations, deeper calf musculature can currently not be assessed in a reproduceable manner, which led to exclusion of 3 patients in total in this study. First steps have been taken to increase reliability of the technology and to achieve standardization across different technology platforms, users and indications[26]. In addition, further patients had to be excluded from the primary analysis, so that the results may only insufficiently reflect a clinical routine collective.

In conclusion, the MSOT-derived saturation in the post-exercise calf muscle was confirmed as a diagnostic biomarker to distinguish between HV and IC patients. This opens further diagnostic possibilities to ensure the ischemic origin of symptoms especially in diagnostic challenging patient cohorts due to various concomitant diseases.

## Data availability

The source data for each figure in this study can be found in the supplementary data files: Supplementary Data contains the data underlying Fig. 2, Fig. 3 and Fig. 4. The full study protocol and reporting summaries are available on the registry of ClinicalTrials.gov (ClinicalTrials.gov Identifier NCT05373927, Protocol Identifier MSOT_IC). The datasets generated and analyzed during the study are available from the corresponding authors on reasonable request. There may be restrictions to data availability due to patient privacy and the General Data Protection Regulation.

## Abbreviations

| | |
|---|---|
| 6MWT | Six-Minute Walk Test |
| ABI | Ankle-brachial-index |
| aTASC | Aggregated TASC II classification |
| a.u | Arbitrary units |
| AUC | Area under the curve |
| CCDS | Color-coded duplex sonography |
| DC | Derivation Cohort |
| Hb | Deoxygenated hemoglobin |
| $HbO_2$ | Oxygenated hemoglobin |
| HV | Healthy volunteer(s) |
| IC | Intermittent claudication |
| MSOT | Multispectral Optoacoustic Tomography |
| $mSO_2$ | MSOT saturation |
| PAD | Peripheral artery disease |
| ROC | Receiver operating characteristic |
| ROI | Region of interest |
| VASCU-QOL-6 | Vascular Quality of Life Questionnaire-6 |
| VC | Validation Cohort |

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

## Acknowledgements

The present work was performed in fulfillment of the requirements for obtaining the degree "Dr. med." (M.C.). U.R. received support from the ELAN Fund at the University Hospital of the FAU Erlangen-Nürnberg. A.R. received funding from Interdisciplinary Center for Clinical Research (IZKF), Junior project J089. F.K. received support from European Research Council under the European Union Horizon H2020 program (ERC Starting Grant No 101115742-IseeG).

## Author contributions

F.K. and U.R. conceived and supervised the study. M.C., J.K., Y.L., J.G., A.T. recruited the patients and performed the investigations the patients. A.P.R., A.L.W., R.R., A.B. provided support for MSOT data analysis. M.U. provided support for angiographic imaging and image analysis. W.U. advised on the statistical analysis plan and final execution on the analyses. W.L., A.M., J.W., C.A.B., M.F.N., M.J.W. provided insight and interpretations of the results. All authors reviewed the manuscript, provided feedback and approved the final version.

## Funding

## Competing interests

The authors declare the following competing interests: A.P.R., M.W., and F.K. are shared patent holders on a related Optoacoustic system described in the study. F.K. and U.R. are advisory board members of iThera Medical GmbH. A.P.R., F.K., and U.R. received travel support from iThera Medical GmbH. A.P.R. and F.K. report travel support and lecture fees from Sanofi Aventis GmbH, Germany. No other conflicts of interest were reported by the remaining authors.

## Additional information

**Milenko Caranovic**[1,2,3], **Julius Kempf** [1,3], **Yi Li** [1], **Adrian P. Regensburger**[4], **Josefine S. Günther** [1,5], **Anna P. Träger**[1,3], **Werner Lang** [1], **Alexander Meyer**[1], **Alexandra L. Wagner**[4], **Joachim Woelfle**[4], **Roman Raming**[4], **Lars-Philip Paulus**[4], **Adrian Buehler** [4], **Wolfgang Uter** [6], **Michael Uder**[2], **Christian-Alexander Behrendt**[7], **Markus F. Neurath** [8,9,10], **Maximilian J. Waldner**[8,9,10], **Ferdinand Knieling** [4,11] ✉ **& Ulrich Rother**[1,11] ✉

[1]Department of Vascular Surgery, University Hospital Erlangen, Friedrich-Alexander-Universität Erlangen-Nürnberg (FAU), Erlangen, Germany. [2]Institute of Radiology, University Hospital Erlangen, Friedrich-Alexander-Universität Erlangen-Nürnberg (FAU), Erlangen, Germany. [3]Faculty of Medicine, Friedrich-Alexander- Universität Erlangen-Nürnberg (FAU), Erlangen, Germany. [4]Department of Pediatrics and Adolescent Medicine, University Hospital Erlangen, Friedrich-Alexander-Universität Erlangen-Nürnberg (FAU), Erlangen, Germany. [5]Department of Vascular and Endovascular Surgery, University Hospital Münster, Westfälische Wilhelm-Universität Münster (WWU), Münster, Germany. [6]Department of Medical Informatics, Biometry and Epidemiology, Friedrich-Alexander-Universität Erlangen-Nürnberg (FAU), Erlangen, Germany. [7]Department of Vascular and Endovascular Surgery, Asklepios Klinik Wandsbek, Asklepios Medical School, Hamburg, Deutschland. [8]Department of Medicine 1, University Hospital Erlangen, Friedrich-Alexander- Universität Erlangen-Nürnberg (FAU), Erlangen, Germany. [9]Deutsches Zentrum für Immuntherapie (DZI), University Hospital Erlangen, Friedrich-Alexander- Universität Erlangen-Nürnberg (FAU), Erlangen, Germany. [10]Erlangen Graduate School in Advanced Optical Technologies (SAOT), Friedrich-Alexander-Universität Erlangen-Nürnberg, Erlangen, Germany. [11]These authors contributed equally: Ferdinand Knieling, Ulrich Rother. ✉e-mail: ferdinand.knieling@uk-erlangen.de; ulrich.rother@uk-erlangen.de

