## [Peer review file · Communications Medicine]

Derivation and validation of a non-invasive optoacoustic imaging biomarker for detection of patients with intermittent claudication

Corresponding Author: Dr Ferdinand Knieling

Version 1:

Reviewer comments:

Reviewer #1

(Remarks to the Author)

The authors have followed a rigorous and correctly designed protocol to investigate the ability of MOST to discriminate patients with severe claudication from healthy controls.

The manuscript is easy to read but I have some significant concerns about the technique and applicability of the results observed.

1/ If the general goal is to test the applicability of the proposed tool to diagnose PAD and claudication, then patients with non-feasible measurement should be considered negative results and should not be excluded from the analysis resulting in decreased sensitivity.

2/ The authors insist on co-morbid condition, and few recent papers have found a significant amount of exercise induced changes in systemic oxygen concentration at exercise in PAD patients. How do the authors account for this?

3/ The delay following exercise is expected to be a major issue in MSOT recording. The authors need to convince the reader that the measurement can be performed within a reasonable period of time after exercise.

4/ The provided images clearly show very heterogeneous distribution of Hb and HbO₂, the authors need to analyze the effect of the area of analysis over the reliability of the data reported.

5/ Last, I am concerned with the ROC analyses for MOST that should be compared to the ROC curve of the other tools. In other words, although the system is clearly an interesting tool for research purpose, I doubt that the data presented are sufficient to support the idea that MOST is of additional value in the diagnostic process of PAD patients.

Finally, I disagree both with the title of the manuscript and with the authors conclusions.

On the one hand, the fact that MOST detected patients suffering claudication does not make the tool to detect "early manifestation" of PAD ... most PAD remain asymptomatic.

On the other hand, the conclusion that MSOT-derived saturation in the post-exercise calf muscle can distinguish between HV and IC patients does not make it a tool to discriminate PAD from various concomitant diseases, specifically when it comes to exercise induced hypoxemia.

Reviewer #2

(Remarks to the Author)

Thank you for this submission and the opportunity to review. I spent several hours on prior publications in an attempt to understand the technology. I will freely admit my understanding is still limited, but enough to feel comfortable with the paper.

The paper is well written and organized.

The groups are well matched.

The

Questions and comments

1. Regarding the depth of muscle tissue, at what depth was the data inaccurate?

2. A table of the abbreviations would be helpful. I had to make my own

3. MSOT change in the post-exercise IC group was the most pronounced and had the least overlap. The data I would like to see in 2 figures and have discussed in text are two-fold.

a) a dot-plot pre/post for each group (Normal and IC) and

b) individual change pre to post with lines and stats on the with average drop in MSOT for each group.

From a clinical use perspective, this is what is needed to create criteria for interpretation.

I would be excited to see this additional analysis.

Thank you.

Reviewer #3

(Remarks to the Author)

This original paper focuses on a very interesting topic of non-invasive ultrasound-based functional imaging method of the calf muscle in patients with PAD.

I have to congratulate the researchers to perform a well-designed and conducted study. It is also well written and structured. I suggest some additional discussion of the following comments:

1. Please specify mSO₂ as MOS saturation and its calculation $HbO_2 / (Hb + HbO_2)$ already in the abstract and method section.
2. As described in the method section all participant performed the heel raise exercise to strain the calf muscle, repeated until pain occur. It would be interesting to know how many times such raise exercise has to be performed in the IC group and in the HV group. Particularly in the HV group it is not clear if they performed such raise exercise until pain occur (as they have no claudication) or was there a fix number of such raise exercise. Was this number different in the IC group and HV group? Were MOST parameters correlated also with the number of raise exercise?
3. In the HV group there were more female (57.7%) compared to the IC group (32%). Were there any differences in the MOST parameter between female and male? Maybe sex will influence also mSO₂.
4. Please also provide information on Hb level in the blood. It would interesting if Hb level in the blood would also affect MOST parameters. Was there any correlation of Hb level in the blood and MOST parameters?
5. It would be also interesting if the differences of ABI before and after exercise also correlates with MOST parameters, particularly with nSO₂.

Reviewer #4

(Remarks to the Author)

Abstract

"n=102 patients were included in two independent derivation (DC) and validation cohorts (VC)" – state explicitly how many in each cohort.

P or p – be consistent.

Methods

"Statistical analyses" section – rename to "Sample size and statistical analyses", or have a separate section for sample size.

"Based on data collected of a previous study cohort, we estimated a minimum sample size of 11 (oxygenated hemoglobin [HbO₂])-33 (deoxygenated hemoglobin [Hb]) participants in each clinical group (HV, IC) (80% power, alpha of 0.05 one-sided)" – All information to recreate the sample size calculation must be provided. What software was used? What test was this based on? What are relevant assumptions? Why was it based on a one-sided test? Justifying the use of a one-sided test is a very high bar – it's only justifiable if the change you're looking for can only go in one direction – that is, it is impossible for a change in the opposite direction to occur. What is the justification for a one-sided test here?

"Single missing values led to exclusion of subjects for the specific analysis." – This is "complete case" analysis.

"In case no muscle was appropriately imaged in the MSOT frames, the subject was excluded from all further analyses." I don't think it's right to say they were excluded – as this is missing data (and this "attrition" should be taken into account in the sample size calculation).

"For all statistical tests p<0.05 was considered statistically significant." Why? p-values should never be reduced to "significant" or "non-significant", as this is too simplistic. P-values need to be interpreted weight of evidence against the relevant null hypothesis on a continuous scale.

"Statistical differences of DC and VC were tested by Fisher test or Welch test" – When was each test used? Presumably "Fisher test" refers to "Fisher's exact test"? "Welch test" (presumably, "Welch's t-test") is appropriate when variances between groups are unequal. Were variances between groups unequal? If so, why? Remember, for a standard t-test,

variables only need to be “approximately” equal. In reality they need to be substantially different to violate the equal variances assumption of a t-test.

“To illustrate the diagnostic quality to differentiate between HV and IC by MSOT, the sensitivity, specificity, area under the curve (AUC) and the cut-off point (derived from the optimal Youden index) were determined in the DC. The cut-off point was then applied to the VC to validate the results from the DC.” – Why was the Youden Index used? Normally in diagnostic studies, there’s some preference for either specificity or sensitivity. Perhaps they are equally important here, but the rationale for maximising both as opposed to favouring one over the other needs to be explained.

Figure 1 (and n for the study generally). The number in each cohort should be given as 59 and 56, not 51/51. State how many are HC/IC in each cohort. The numbers “excluded” are missing data. Rather than calling them excluded, change the lower boxes to indicate “n Included in primary analysis” and “n not included in primary analysis” (with reasons listed below). While use of the word “excluded” is not technically wrong, these participants are not “excluded” in the same way as the 8 not giving informed consent are excluded. The “not included in primary analysis” participants should still be analysed as far as possible – and should be included in participant demographics etc.

Results

The words “significant” and “non-significant” are used frequently. It’s better to remove this kind of language entirely. Instead, provide estimates, 95% Confidence intervals and p-values. But don’t make interpretation contingent upon any arbitrary threshold (i.e. 0.05). Interpret differences based on their biological (not statistical) importance. If you want to say there’s a big difference, don’t say “significant” as this makes people think of statistical tests – use a different word, like “substantial”.

Correlation to clinical standard assessments

There are a lot of correlations shown, but I’m not sure exactly what the objective was here. Correlations only provide limited information – would it make sense to additionally examine bias and agreement for any of these relationships? (i.e. Bland-Altman plots).

Funding

“Fond” -> Fund.

General comments

There is a Statistical Analysis Plan included, but it isn’t mentioned in the paper, so I only realised this after reviewing the paper. It’s good that one exists, but there’s a lot in the paper that doesn’t match what is in the SAP. The sample size details are quite different in the paper. Please ensure that all the analyses presented in the paper match what is written in the SAP – there is a very extensive list of secondary endpoints.

SAP states analyses were conducted in SPSS and R, main text says SPSS and GraphPad.

Was a statistician involved in writing the SAP and paper?

Inclusion and exclusion criteria need to be stated in the main text.

Version 2:

Reviewer comments:

Reviewer #1

(Remarks to the Author)

Thank you for your answers to the comments.

I have no further suggestions

Reviewer #3

(Remarks to the Author)

The authors addressed the reviewer’s comments and changed the manuscript accordingly. The manuscript substantially improved. No further comments are required.

Reviewer #4

(Remarks to the Author)

I have some further comments on the revised manuscript:

Use of word "significant" has been reduced throughout the text, but the concept of "statistical significance" is still used (The methods states "For all statistical tests $p < 0.05$ was considered statistically significant"). P-values should not be interpreted around any arbitrary threshold - they represent weight of evidence against a null hypothesis and should be interpreted on a continuous scale. To this end, it is important that if p-values are given in text or in graphs, p-values for ALL tests should be given. None should be omitted or labelled "n.s." because they are "not significant". P-values still have meaning above 0.05 (0.061 is very different to 0.610, for example).

If there are too many p-values to show on graphs, give the comparisons and corresponding estimates, 95% CIs and p-values

in separate tables.

Reviewer 1 and I have asked for slightly different, mutually exclusive, things regarding missing data. Reviewer 1 ask for those with missing data to be included, I asked for participants with missing data to be included as far as possible *i.e. in demographics) but not in the analyses where data was missing. As it stands, ROC curves have been repeated with assumptions made about missing data, but the demographic tables only include those without missing data. Repeating analyses with different assumptions is a valid approach generally (sensitivity analysis) as gives an indication of the effect of assumptions/decisions made in the primary analysis. However, I don't think including those with missing data, by making assumptions about that data, is the right approach here - particularly when assumptions are simplistic (like setting missing values to means). The number/percentage of participants that cannot be assessed is an important result in itself. I don't think the new figure 4 should be included - rather, the ROC curves from the primary analysis should be interpreted in the context that not this technique cannot be applied to all patients.

Correlations: The authors haven't justified the use of simple correlations to me - essentially saying the these correlation are useful to clinicians. However, the limitations of correlations like these are typically not appreciated by clinicians. This section either needs to be extended using the Bland-Altman approach to further assess agreement and bias, or these correlations need to be interpreted with explicit consideration of their limitations.

It may also be beneficial to perform some modelling analysis to look at adjusted relationships between outcomes (e.g. adjusting for gender and age).

Minor points:

Page 10 line 277 typo: "It i possible to distinguish between HC..."

Reviewer #1

The authors have followed a rigorous and correctly designed protocols to investigate the ability of MOST to discriminate patients with severe claudication from healthy controls. The manuscript is easy to read but I have some significant concerns about the technique and applicability of the results observed.

Remark 1

If the general goal is to test the applicability of the proposed tool to diagnose PAD and claudication, then patients with non-feasible measurement should be considered negative results and should not be excluded from the analysis resulting in decreased sensitivity.

Response 1:

We thank the reviewer for the very valuable suggestion. To perform an analysis, we kept the original assigned of each participant and have made to different assumption:

1. Mean of all patients: All participants with missing MSOT values were assigned an inconclusive result, i.e. the mean of all measured values (healthy volunteers and intermitted claudication).

2. Mean per group: All participants with missing MSOT values were assigned an inconclusive result within their study group, i.e. the mean of HV OR IC group.

The results are presented in **Figure 4**.

Remark 2

The authors insist on co-morbid condition, and few recent papers have found a significant amount of exercise induced changes in systemic oxygen concentration at exercise in PAD patients. How do the authors account for this?

Response 2

We were able to recognize 2 aspects of the described differences in the MSOT parameters of the calf muscles. Firstly, there are differences depending on the stage of the disease () and secondly, the load leads to a dynamic

change. As an example, we show this 'reperfusion' after exercise in Figure xx. As can be seen from the matched data, this effect is diametrically different in almost all healthy and diseased patients.

Matched comparison (absolute)

Matched comparison (relative)

Gardner AW, Montgomery PS, Wang M, Liang M. Effects of Long-Term Home Exercise in Participants With Peripheral Artery Disease. *J Am Heart Assoc.* 2023 Nov 7;12(21):e029755. doi: 10.1161/JAHA.122.029755. Epub 2023 Nov 6. PMID: 37929770; PMCID: PMC10727372.

Li Z, Englund EK, Langham MC, Feng J, Jia K, Floyd TF, Yodh AG, Baker WB. Exercise Training Increases Resting Calf Muscle Oxygen Metabolism in Patients with Peripheral Artery Disease. *Metabolites.* 2021 Nov 29;11(12):814. doi: 10.3390/metabo11120814. PMID: 34940572; PMCID: PMC8706023.

Remark 3

The delay following exercise is expected to be a major issue in MSOT recording. The authors need to convince the reader that the measurement can be performed within a reasonable period of time after exercise.

Response 3

Thank you for your question. As we also expected time to be a major issue in this patient cohort, we analyzed the individual reperfusion profiles in our included cohort. As one can expect the reperfusion profiles between HVs and ICs are different. Therefore, the optimal time period to assess the patients after provocation was between 150 and 250 seconds after provocation (compare figure attached). This appears to be a useful time interval in which most of the claudicants should be assessable despite their age. However, as demonstrated with Figure 3, the interval of reperfusion remains changed for >500s, which provides enough time for imaging.

Remark 4

The provided images clearly show very heterogeneous distribution of Hb and HbO₂, the authors need to analyze the effect of the area of analysis over the reliability of the data reported.

Response 4

Thank you for your important question. As you are absolutely right, muscle perfusion appears to be heterogeneous over the muscle. First, individual vessels may be cut into the image, which under certain circumstances may have a higher oxygenation level, which can falsify the calculation (averaging). Therefore, each of our regions of interest was set manually to avoid vessels being included. Secondly, all our values were calculated as averages (over 7 frames) out of a ROI, therefore the heterogeneity was outweighed until a certain degree. However, the concerns expressed can only be completely dispelled in a separate study. We are currently conducting a study in which a three-dimensional recording of the calf muscle is made over a distance of 5 cm in both healthy and PAD patients. In addition, the measurements are also being carried out in a 2d plane. this study is the only way to provide a conclusive answer to the question.

However, the degree of heterogeneity in different muscles has been studied before in healthy subjects.

Please refer to Figure 6 in our prior work:

Wagner AL, Danko V, Federle A, Klett D, Simon D, Heiss R, Jüngert J, Uder M, Schett G, Neurath MF, Woelfle J, Waldner MJ, Trollmann R, Regensburger AP, Knieling F. Precision of handheld multispectral optoacoustic tomography for muscle imaging. *Photoacoustics*. 2020 Nov 11;21:100220. doi: 10.1016/j.pacs.2020.100220. PMID: 33318928; PMCID: PMC7723806.

For HbT a variation of 5.4% (depending on a different position on the muscle) was found.

Remark 5

Last, I am concerned with the ROC analyses for MOST that should be compared to the ROC curve of the other tools. In other words, although the system is clearly an interesting tool for research purpose, I doubt that the data presented are sufficient to support the idea that MOST is of additional value in the diagnostic process of PAD patients.

Response 5

Thank you for your question. We absolutely agree, that the true impact of MSOT in PAD diagnosis has to be shown in future. However, MSOT is the first tool, despite maybe MRA imaging, that can noninvasively measure saturation values and perfusion values on the level of musculature in depth. From a clinical perspective we can imagine that this might be of particular relevance of different clinical scenarios. First, patients doing structured exercise therapy (level IA recommendation for claudicants ESVS Guideline) can be monitored despite the walking distance objectively by MSOT. This can be a motivation to continue training if you see the improvement but do not yet perceive it subjectively. Second, MSOT seems not to be influenced by the diagnostic gap, resulting from mediasclerosis in the ankle-brachial index measurement. Results attached from our first study demonstrate, that MSOT values are not influenced from the media vessel calcification:

ABI

Guenther et al, JACC Cardiovasc Imaging doi: 10.1016/j.jcmg.2022.11.010

Against this background, the aim of this study was initially to investigate whether a valid differentiation between healthy subjects and patients with claudication is possible. However, the evaluation of the value of the method in future PAD diagnostics must be investigated in further studies, as well as the evaluation of the methodology compared to other diagnostic methods.

Remark 6

Finally, I disagree both with the title of the manuscript and with the authors conclusions.

On the one hand, the fact that MOST detected patients suffering claudication does not make the tool to detect "early manifestation » of PAD ... most PAD remain asymptomatic.

On the other hand, the conclusion that MSOT-derived saturation in the post-exercise calf muscle can distinguish between HV and IC patients does not make it a tool to discriminate PAD from various concomitant diseases, specifically when it comes to exercise induced hypoxemia.

Response 6

Thank you for these comments. We absolutely agree that the title and the conclusion were rather misleading.

Therefore, we changed the title to:

"Derivation and validation of a non-invasive optoacoustic imaging biomarker for detection of patients with intermittent claudication"

The conclusion as also adapted to:

"In conclusion, the MSOT-derived saturation in the post-exercise calf muscle was confirmed as a new diagnostic biomarker to distinguish between HV and IC patients. This might open new diagnostic possibilities in future PAD diagnostic, which, however, has to be shown in further studies."

Reviewer #2 (Remarks to the Author):

Thank you for this submission and the opportunity to review. I spent several hours on prior publications in an attempt to understand the technology. I will freely admit my understanding is still limited, but enough to feel comfortable with the paper.

The paper is well written and organized.
The groups are well matched.

Questions and comments

Remark 1

Regarding the depth of muscle tissue, at what depth was the data inaccurate?

Response 1

This correlation was not explicitly investigated in the study. On the other hand, no patient was primarily excluded as part of the screening, so that we obtain a more realistic patient cohort. Overall, however, it must be noted that the gastrocnemius muscle can almost always be reached in the range of 0.5-1.5 cm.

The dependence of depth and signal was studied before:

Wagner AL, Danko V, Federle A, Klett D, Simon D, Heiss R, Jüngert J, Uder M, Schett G, Neurath MF, Woelfle J, Waldner MJ, Trollmann R, Regensburger AP, Knieling F. Precision of handheld multispectral optoacoustic tomography for muscle imaging. *Photoacoustics*. 2020 Nov 11;21:100220. doi: 10.1016/j.pacs.2020.100220. PMID: 33318928; PMCID: PMC7723806.

However, in prior studies, we found that a single wavelength approach may be less prone to depth related inaccuracies:

Träger AP, Günther JS, Raming R, Paulus LP, Lang W, Meyer A, Kempf J, Caranovic M, Li Y, Wagner AL, Tan L, Danko V, Trollmann R, Woelfle J, Klett D, Neurath MF, Regensburger AP, Eckstein M, Uter W, Uder M, Herrmann Y, Waldner MJ, Knieling F, Rother U. Hybrid ultrasound and single wavelength optoacoustic imaging reveals muscle degeneration in peripheral artery disease. *Photoacoustics*. 2023 Dec 2;35:100579. doi: 10.1016/j.pacs.2023.100579. PMID: 38312805; PMCID: PMC10835356.

Remark 2

A table of the abbreviations would be helpful. I had to make my own

Response 2

Thank you for this suggestion, a table is included and updated.

Remark 3

MSOT change in the post-exercise IC group was the most pronounced and had the least overlap. The data I would like to see in 2 figures and have discussed in test are two-fold.

a) a dot-plot pre/post for each group (Normal and IC) and

b) individual change pre to post with lines and stats on the with average drop in MSOT for each group.

Response 3

As suggested, we calculated both (a) and (b) aspects. The data is now included in Figure 2.

Matched comparison (absolute)

Matched comparison (relative)

Reviewer #3 (Remarks to the Author):

This original paper focuses on a very interesting topic of non-invasive ultrasound-based functional imaging method of the calf muscle in patients with PAD.

I have to congratulate the researchers to perform a well-designed and conducted study. It is also well written and structured.

I suggest some additional discussion of the following comments:

Remark 3-1

1. Please specify mSO₂ as MOS saturation and its calculation $HbO_2 / (Hb+HbO_2)$ already in the abstract and method section.

Response 3-1

Thank you for the suggestion. We included its calculation already in the abstract and in the methods section.

Remark 3-2

2. As described in the method section all participant performed the heel raise exercise to strain the calf muscle, repeated until pain occur. It would be interesting to know how many times such raise exercise has to be performed in the IC group and in the HV group. Particularly in the HV group it is not clear if they performed such raise exercise until pain occur (as they have no claudication) or was there a fix number of such raise exercise. Was this number different in the IC group and HV group? Were MOST parameters correlated also with the number of raise exercise?

Response 3-2

Thank you for this interesting question. Unfortunately, we did not count the numbers of heel raises done by each participant. However, even in the healthy cohort all heel raises were done until exhaustion occurred. That was achieved in each of the participant, as this kind of provocation appears to be very exhaustive even in non-claudicants. However, we measured how long heel raises were conducted in seconds. By comparing this, the mean duration of heel raises in the claudicant group was 84.4 (SD 28.5) seconds and in the healthy group 129.9 (SD 52.2) seconds. We specified this more in detail in the methods section and **Table 1**.

Remark 3-3

3. In the HV group there were more female (57.7%) compared to the IC group (32%). Were there any differences in the MOST parameter between female and male? Maybe sex will influence also mSO₂.

Response 3-3

We thank the reviewer for this important suggestion. Interestingly the gender has a influence on gender (as described earlier: Wagner AL, Danko V, Federle A, Klett D, Simon D, Heiss R, Jüngert J, Uder M, Schett G, Neurath MF, Woelfle J, Waldner MJ, Trollmann R, Regensburger AP, Knieling F. Precision of handheld multispectral optoacoustic tomography for muscle imaging. Photoacoustics. 2020 Nov 11;21:100220. doi: 10.1016/j.pacs.2020.100220. PMID: 33318928; PMCID: PMC7723806.).

Moreover, mSO₂ is not biased by a gender effect. The data is included in the **Supplementary Figure S7**.

Remark 3-4

4. Please also provide information on Hb level in the blood. It would be interesting if Hb level in the blood would also affect MOST parameters. Was there any correlation of Hb level in the blood and MOST parameters?

Response 3-4

We thank the reviewer for this question. This is a very interesting point. As suggested, we correlated the Hb levels and MSOT parameters. The data is now included in **Supplementary Figure S6**. We did not find strong evidence that there is a correlation of Hb levels and MSOT parameters in muscle measurements. However, there was a study showing that detection of severe anemia is feasible. In contrast to our work, these measurements were not taken on muscle (which may present a broader absorbing tissue containing myo- and hemoglobin).

Reference

Ganzleben I, Klett D, Hartz W, Götzfried L, Vitali F, Neurath MF, Waldner MJ. Multispectral optoacoustic tomography for the non-invasive identification of patients with severe anemia in vivo. *Photoacoustics*. 2022 Oct 12;28:100414. doi: 10.1016/j.pacs.2022.100414. PMID: 36276233; PMCID: PMC9583176.

Remark 3-5

5. It would be also interesting if the differences of ABI before and after exercise also correlates with MOST parameters, particularly with nSO₂.

Response 3-5

As suggested, we correlated the differences in ABI (Δ ABI = postABI - preABI) and MSOT parameters. Subjects with a pathologic postABI may also have lower MSOT parameters (HbO₂ and msO₂).

Reviewer #4 (Remarks to the Author):

Remark 4-1:

Abstract

"n=102 patients were included in two independent derivation (DC) and validation cohorts (VC)" – state explicitly how many in each cohort.

Response 4-1:

The abstract was changed as requested.

Remark 4-2:

P or p – be consistent

Response 4-2:

The nomenclature was changed to a consistent format.

Remark 4-3:

Methods

"Statistical analyses" section – rename to "Sample size and statistical analyses", or have a separate section for sample size.

Response 4-3:

The sections were separated.

Remark 4-4:

"Based on data collected of a previous study cohort, we estimated a minimum sample size of 11 (oxygenated hemoglobin [HbO₂]-33 (deoxygenated hemoglobin [Hb]) participants in each clinical group (HV, IC) (80% power, alpha of 0.05 one-sided)" – All information to recreate the sample size calculation must be provided. What software was used? What test was this based on? What are relevant assumptions? Why was it based on a one-sided test? Justifying the use of a one-sided test is a very high bar – it's only justifiable if the change you're looking for can only go in one direction – that is, it is impossible for a change in the opposite direction to occur. What is the justification for a one-sided test here?

Response 4-4:

Thank you for the question, giving us the possibility to clarify this issue (Please compare also the SAP). Statistical sample size calculation based on the results of the previous study (MSOT_PAD, NCT04641091) suggests a necessary study collective of at least 48 to a maximum of 124 subjects/patients. The two-sample t-test power calculation with a power of 0.8 gave an n of about 12 for each group (study group 1 - healthy volunteers, study group 1 – claudication patients, study group 2 - healthy volunteers, study group 2 - claudication patients). Additional sample size planning based on the AUC, which should be at least 0.7 for a relevant significance, gave an n of 31 subjects/patients for each group (with a significance level of 0.05 and a power of 0.8).

These calculations were based on data comparing healthy subjects with patients in stage IIb who, however, had not been adequately exercised (all only with the same walking distance of 150 meters). Due to the hypothesis of the study that the ability to differentiate between healthy subjects and claudication patients can be increased by adequate provocation testing, the sample size estimated here on the basis of the previous data is to be regarded as conservative. For this reason, after the inclusion of the first 25 patients and the first 25 healthy test persons, the necessary total sample size will be reevaluated in cooperation with the Institute for Biometry – on the basis of the data collected up to that point. During this interim analysis we found, a minimum sample size of 11 (oxygenated hemoglobin [HbO₂]-33 (deoxygenated hemoglobin [Hb]) participants in each clinical group (HV, IC) (80% power, alpha of 0.05, two-sample t-test power calculation two-sided). We apologize for the unclear and in parts incorrect description in the former version of the manuscript.

Remark 4-5:

"Single missing values led to exclusion of subjects for the specific analysis." – This is "complete case" analysis.

Response 4-5:

Nomenclature was corrected.

Remark 4-6:

"In case no muscle was appropriately imaged in the MSOT frames, the subject was excluded from all further analyses." I don't think it's right to say they were excluded – as this is missing data (and this "attrition" should be taken into account in the sample size calculation).

Response 4-7:

Thank you for this comment. As we were aware of the missing data, we chose a larger group of patients for each group than the minimum number identified in our sample size calculation. As mentioned above, this calculation resulted in a requirement of 11 patients in each group. However, we ended up with approximately 25 patients in each group.

Remark 4-8:

"For all statistical tests $p < 0.05$ was considered statistically significant." Why? p-values should never be reduced to "significant" or "non-significant", as this is too simplistic. P-values need to be interpreted weight of evidence against the relevant null hypothesis on a continuous scale.

Response 4-8:

Wording was specified.

Remark 4-9:

"Statistical differences of DC and VC were tested by Fisher test or Welch test" – When was each test used? Presumably "Fisher test" refers to "Fisher's exact test"? "Welch test" (presumably, "Welch's t-test") is appropriate when variances between groups are unequal. Were variances between groups unequal? If so, why? Remember, for a standard t-test, variables only need to be "approximately" equal. In reality they need to be substantially different to violate the equal variances assumption of a t-test.

Response 4-9:

We excuse for the error. All test were reviewed and were already performed as t-test. The passage was corrected.

Remark 4-10:

"To illustrate the diagnostic quality to differentiate between HV and IC by MSOT, the sensitivity, specificity, area under the curve (AUC) and the cut-off point (derived from the optimal Youden index) were determined in the DC. The cut-off point was then applied to the VC to validate the results from the DC." – Why was the Youden Index used? Normally in diagnostic studies, there's some preference for either specificity or sensitivity. Perhaps they are equally important here, but the rationale for maximising both as opposed to favouring one over the other needs to be explained.

Response 4-10:

Youden's index was chosen to maximize the sum of sensitivity and specificity, because (given prior studies) we suspected AUC is high (eg, >0.80).

Mascha, Edward J. PhD. Identifying the Best Cut-Point for a Biomarker, or Not. *Anesthesia & Analgesia* 127(4):p 820-822, October 2018. | DOI: 10.1213/ANE.0000000000003680

Remark 4-11:

Figure 1 (and n for the study generally). The number in each cohort should be given as 59 and 56, not 51/51. State how many are HC/IC in each cohort. The numbers "excluded" are missing data. Rather than calling them excluded, change the lower boxes to indicate "n Included in primary analysis" and "n not included in primary analysis" (with reasons listed below). While use of the word "excluded" is not technically wrong, these participants are not "excluded" in the same way as the 8 not giving informed consent are excluded. The "not included in primary analysis" participants should still be analysed as far as possible – and should be included in participant demographics etc.

Response 4-11:

We adapted the flowchart as proposed. Analyses for the entire group was included as proposed by reviewer #1 (intention-treat-analyses).

Remark 4-12:

Results

The words “significant” and “non-significant” are used frequently. It’s better to remove this kind of language entirely. Instead, provide estimates, 95% Confidence intervals and p-values. But don’t make interpretation contingent upon any arbitrary threshold (i.e. 0.05). Interpret differences based on their biological (not statistical) importance. If you want to say there’s a big difference, don’t say “significant” as this makes people think of statistical tests – use a different word, like “substantial”.

Response 4-12:

The language was corrected throughout.

Remark 4-13:

Correlation to clinical standard assessments

There are a lot of correlations shown, but I’m not sure exactly what the objective was here. Correlations only provide limited information – would it make sense to additionally examine bias and agreement for any of these relationships? (I.e. Bland-Altman plots).

Response 4-13:

We would like to thank the reviewer for this suggestion. Standard clinical examinations are used to assess the patient's condition (in various aspects). In doing so, completely independent other functions are tested (e.g. walking distance, personal well-being, angiographic status of the vasculature) which, although related to the disease, represent a completely different measurement category. For the clinician, however, a correlation between novel measured values and the standard is a very helpful statement.

Remark 4-14:

Funding

“Fond” -> Fund.

Response 4-14:

Corrected

Remark 4-15:

General comments

There is a Statistical Analysis Plan included, but it isn’t mentioned in the paper, so I only realised this after reviewing the paper. It’s good that one exists, but there’s a lot in the paper that doesn’t match what is in the SAP. The sample size details are quite different in the paper. Please ensure that all the analyses presented in the paper match what is written in the SAP – there is a very extensive list of secondary endpoints.

SAP states analyses were conducted in SPSS and R, main text says SPSS and GraphPad.

Response 4-15:

Thank you for the comment. It is correct that the programs SPSS and R should be used for statistical analysis when planning the study. In the course of the statistical analysis, one of the co-authors created some graphs using Graph, as this program was particularly suitable for the presentation of our data. Overall, however, the data was primarily analyzed using SPSS.

Remark 4-16:

Was a statistician involved in writing the SAP and paper?

Response 4-16:

Yes, Prof. Dr. med. Wolfgang Uter.

Remark 4-17:

Inclusion and exclusion criteria need to be stated in the main text.

Response 4-17:

The inclusion and exclusion criteria were added.

Point-to-point reply

Reviewer #4

Comment 1:

Use of word "significant" has been reduced throughout the text, but the concept of "statistical significance" is still used (The methods states "For all statistical tests $p < 0.05$ was considered statistically significant"). P-values should not be interpreted around any arbitrary threshold - they represent weight of evidence against a null hypothesis and should be interpreted on a continuous scale. To this end, it is important that if p-values are given in text or in graphs, p-values for ALL tests should be given. None should be omitted or labelled "n.s." because they are "not significant". P-values still have meaning above 0.05 (0.061 is very different to 0.610, for example).

If there are too many p-values to show on graphs, give the comparisons and corresponding estimates, 95% CIs and p-values in separate tables.

Response 1:

We included all missing p-values and rewrote the section as follows:

The threshold for statistical significance was set to a p-value of < 0.05 . When interpreting differences between groups, the p-value, effect size and overall consistency of the results were considered.

Comment 2:

Reviewer 1 and I have asked for slightly different, mutually exclusive, things regarding missing data. Reviewer 1 ask for those with missing data to be included, I asked for participants with missing data to be included as far as possible *i.e. in demographics) but not in the analyses where data was missing. As it stands, ROC curves have been repeated with assumptions made about missing data, but the demographic tables only include those without missing data.

Repeating analyses with different assumptions is a valid approach generally (sensitivity analysis) as gives an indication of the effect of assumptions/decisions made in the primary analysis. However, I don't think including those with missing data, by making assumptions about that data, is the right approach here - particularly when assumptions are simplistic (like setting missing values to means). The number/percentage of participants that cannot be assessed is an important result in itself.

I don't think the new figure 4 should be included - rather, the ROC curves from the primary analysis should be interpreted in the context that not this technique cannot be applied to all patients.

Response 2:

As proposed, we removed the requested analysis from reviewer 1 and included extended demographics as requested.

We therefore extended the section "Study limitations":

In addition, further patients had to be excluded from the primary analysis, so that the results may only insufficiently reflect a clinical routine collective.

Comment 3:

Correlations: The authors haven't justified the use of simple correlations to me - essentially saying the these correlation are useful to clinicians. However, the limitations of correlations like these are typically not appreciated by clinicians. This section either needs to be extended using the Bland-Altman approach to further assess agreement and bias, or these correlations need to be interpreted with explicit consideration of their limitations.

It may also be beneficial to perform some modelling analysis to look at adjusted relationships between outcomes (e.g. adjusting for gender and age).

Response 3

As proposed, we decided to include Bland-Altman analyses. This is now included in the new **Figure 4**.

Figure 4 – Correlation of MSOT parameters with clinical assessments

Correlation matrix of the MSOT parameters deoxygenated hemoglobin (Hb), oxygenated hemoglobin (HbO₂), MSOT saturation (mSO₂) before (Pre) and after (Post) heel raise exercise with ankle-brachial-index after heel raise exercise (ABI), the absolute walking distance in the Six-Minute Walk Test (6MWT), the score in the Vascular Quality of Life Questionnaire-6 (VASCUQOL-6), the score in the aggregated Trans-Atlantic Inter-Society Consensus Document on Management of Peripheral Arterial Disease classification II (aTASC). Correlation given by Pearson correlation coefficient (R) are indicated as strongly positive (dark blue) and strongly negative (dark red) (**A**). Bland-Altman plots of z-scores for Hb (**B**), HbO₂ (**C**) and mSO₂ (**D**) in relation to ABI, 6MWT, VASCUQOL-6 and aTASC. The solid orange line represents the mean difference of both values, while the black dashed lines represent the mean difference (± 1.96 SD).

However, we decided against re-analysing gender-specific differences. this was a) already addressed by the questions in the previous review and b) comprehensively investigated in other studies.

Wagner AL, Danko V, Federle A, Klett D, Simon D, Heiss R, Jüngert J, Uder M, Schett G, Neurath MF, Woelfle J, Waldner MJ, Trollmann R, Regensburger AP, Knieling F. Precision of handheld multispectral optoacoustic tomography for muscle imaging. *Photoacoustics*. 2020 Nov 11;21:100220. doi: 10.1016/j.pacs.2020.100220. PMID: 33318928; PMCID: PMC7723806.

Träger AP, Günther JS, Raming R, Paulus LP, Lang W, Meyer A, Kempf J, Caranovic M, Li Y, Wagner AL, Tan L, Danko V, Trollmann R, Woelfle J, Klett D, Neurath MF, Regensburger AP, Eckstein M, Uter W, Uder M, Herrmann Y, Waldner MJ, Knieling F, Rother U. Hybrid ultrasound and single wavelength optoacoustic imaging reveals muscle degeneration in peripheral artery disease. *Photoacoustics*. 2023 Dec 2;35:100579. doi: 10.1016/j.pacs.2023.100579. PMID: 38312805; PMCID: PMC10835356.

Comment 4

Minor points:

Comment x

Page 10 line 277 typo: "It i possible to distinguish between HC..."

Response 4

Typo corrected.

Comment 5

Additional requests from Reviewer 4:

I think the authors have addressed reviewer 2's comments, for the most part, however there are issues with the graphs they have included – Figure 2 panels D and E.

In panel D:

The bars serve no purpose so should be removed (individual points with error bars convey all the information).

Error bars and mean should be plotted on top of the individual points instead of underneath.

In the relative comparison, the 'pre' bars should be removed, as they just take the value 1 with no variation around them.

The 'post' points should be labelled as difference (post-pre).

Removing these the pre bars will remove the statistical comparison between the two relative 'pre' groups – the two groups with the value 1 and no variability – which is good because this comparison is entirely nonsensical.

I think it would actually make most sense for them to drop panel D entirely, since panels D and E essentially show the same thing. Panel E does it better (and keeping the 'pre' bars on the relative graph is not so bad here). If they want to show between group comparisons, they could be added here. As long as they don't compare HV (pre) and IC (pre) on the relative graph.

Response 5

As proposed by the reviewer, we removed the panel D and adjusted panel E.

Figure 2 – MSOT images of a HV and an IC before and after the heel raise exercise
MSOT Imaging approach in the study (A). Dot plots of all patients in the derivation cohort pre- and post-exercise (B). MSOT images of a healthy volunteer (HV) and a patient with

peripheral arterial disease in Fontaine stage IIa/IIb or Rutherford category 1 to 3 (IC) before (Pre) and after (Post) the heel raise exercise showing the parameters deoxygenated hemoglobin (Hb), oxygenated hemoglobin (HbO₂) and MSOT saturation (mSO₂). The first column shows a schematic representation of the visible anatomical structures (skin line, subcutaneous fat, muscle); the second shows the sonographic image in B-mode. The following columns illustrate the MSOT data of Hb, HbO₂ and mSO₂, visually represented as a heat map, with the colors reflecting the signal intensity (Hb: dark blue, minimum signal; yellow, maximum signal; HbO₂: dark red, minimum signal; yellow, maximum signal; mSO₂: dark blue, minimum signal; dark red, maximum signal) (C). Individual comparison of HV and IC patients using absolute (E, right) or relative mSO₂ measurements (left). Lines connect individual patients. Figure created with bioRender.com

Comment 6

Finally, they need to check the p-values presented. As far as I can see, the comparisons made in panels D and E should be the same – i.e. within participant differences (+between group differences in panel D), but the p-values are different, so something is wrong here. I think either the p-values have been reported incorrectly, or panel D shows inappropriate non-paired comparisons for within participant differences (if the latter is the case, more reason to drop panel D entirely).

Response 6

We thank the reviewer for the comment and excuse for the error.

As proposed in Comment 5, we dropped Panel D in Figure 2 and checked on the analyses/p-values shown. See also **Response 5**.

Point-to-point reply

Reviewer #4

Comment 1:

The reviewer queried Figure 4, as they do not think enough information is given to immediately understand the Bland-Altman plots, such as the choice of colours and the presence of diagonal lines. They think it would be best to remove these plots and just present the simple correlations.

Response 1:

The content was removed as requested from the figure and text.